# Replication Study: Discovery and preclinical validation of drug indications using compendia of public gene expression data

**Irawati Kandela, Fraser Aird, Reproducibility Project: Cancer Biology***

Developmental Therapeutics Core, Northwestern University, Evanston, United States

**\*For correspondence:** tim@cos.io; nicole@scienceexchange.com

**Group author details:**
Reproducibility Project: Cancer Biology See page 9

**Abstract** In 2015, as part of the Reproducibility Project: Cancer Biology, we published a Registered Report (Kandela et al., 2015) that described how we intended to replicate selected experiments from the paper "Discovery and Preclinical Validation of Drug Indications Using Compendia of Public Gene Expression Data" (Sirota et al., 2011). Here we report the results of those experiments. We found that cimetidine treatment in a xenograft model using A549 lung adenocarcinoma cells resulted in decreased tumor volume compared to vehicle control; however, while the effect was in the same direction as the original study (Figure 4C; Sirota et al., 2011), it was not statistically significant. Cimetidine treatment in a xenograft model using ACHN renal cell carcinoma cells did not differ from vehicle control treatment, similar to the original study (Supplemental Figure 1; Sirota et al., 2011). Doxorubicin treatment in a xenograft model using A549 lung adenocarcinoma cells did not result in a statistically significant difference compared to vehicle control despite tumor volume being reduced to levels similar to those reported in the original study (Figure 4C; Sirota et al., 2011). Finally, we report a random effects meta-analysis for each result. These meta-analyses show that the inhibition of A549 derived tumors by cimetidine resulted in a statistically significant effect, as did the inhibition of A549 derived tumors by doxorubicin. The effect of cimetidine on ACHN derived tumors was not statistically significant, as predicted.

## Introduction

The Reproducibility Project: Cancer Biology (RP:CB) is a collaboration between the Center for Open Science and Science Exchange that seeks to address concerns about reproducibility in scientific research by conducting replications of selected experiments from a number of high-profile papers in the field of cancer biology (*Errington et al., 2014*). For each of these papers a Registered Report detailing the proposed experimental designs and protocols for the replications was peer reviewed and published prior to data collection. The present paper is a Replication Study that reports the results of the replication experiments detailed in the Registered Report (*Kandela et al., 2015*) for a paper by Sirota et al., and uses a number of approaches to compare the outcomes of the original experiments and the replications.

In 2011, Sirota et al. presented a computational approach to identify drugs that could be repurposed for alternative diseases. Cimetidine, an H2 receptor blocker used to treat gastric ulcers (*Kubecova et al., 2011*), was predicted to be a therapeutic approach for lung adenocarcinoma. Using a xenograft model with the A549 lung adenocarcinoma cell line, cimetidine resulted in a statistically significant decrease in tumor volume on the last day of the experiment compared to vehicle

control (*Sirota et al., 2011*). A decrease in tumor volume was also observed with the chemotherapeutic agent doxorubicin, a positive control therapy. Cimetidine treatment was also tested in mice implanted with the ACHN renal cell carcinoma cell line, which did not change tumor volume compared to vehicle control. This was in agreement with the prediction and demonstrated specificity of the approach by Sirota and colleagues (*Sirota et al., 2011*).

The Registered Report for the paper by Sirota et al. described the experiments to be replicated (Figure 4C–D and Supplemental Figure 1), and summarized the current evidence for these findings (*Kandela et al., 2015*). The outcome measures reported in this Replication Study will be aggregated with those from the other Replication Studies to create a dataset that will be examined to provide evidence about reproducibility of cancer biology research, and to identify factors that influence reproducibility more generally.

## Results and discussion

### Assessing the effect of cimetidine treatment on tumor growth in a xenograft model of lung carcinoma and a xenograft model of renal carcinoma

We sought to independently replicate the effectiveness of cimetidine treatment as a possible therapeutic agent for lung adenocarcinoma, but not renal cell carcinoma, by performing an experiment in mice implanted with the A549 or ACHN cell lines, respectively. This experiment is similar to what was reported in Figure 4C–D and Supplemental Figure 1 of *Sirota et al., 2011*. Tumor volume was determined by caliper daily for the duration of the study (*Figure 1*, *Figure 1—figure supplement 1*). While the original study included three doses of cimetidine in the experimental design, this replication attempt was restricted to only the highest dose (100 mg/kg). Tumors from A549 cells treated with cimetidine grew to an average of 2.37 times their original volume by day 11 [n=13, *SD*=0.723], similar to the ~2.3 times reported in the original study. This is in comparison to tumors from A549 cells treated with vehicle, which grew to an average of 3.29 times their original volume by day 11 [n=14, *SD*=1.17], similar to the ~3.3 times reported in the original study. Tumors from ACHN cells treated with vehicle or cimetidine grew to an average of 2.96 [n=15, *SD*=0.952] or 3.37 [n=15, *SD*=1.50] times their original volume by day 11, respectively. ACHN derived tumors were similar in volume compared to tumors from A549 cells treated with vehicle in this study. This differed from the tumor volumes reported in *Sirota et al., 2011*, which were ~2.1 and ~2.0 times the original volume in tumors from ACHN cells treated with vehicle or cimetidine, respectively, compared to ~3.3 times the original volume in tumors from A549 cells treated with vehicle. Additionally, the relative standard deviation (RSD) associated with this replication attempt (A549 – vehicle treated = 35.6%; A549 – cimetidine treated = 30.5%; ACHN – vehicle treated = 32.1%; ACHN – cimetidine treated = 44.5%) was generally larger than the RSD observed in the original study (A549 – vehicle treated = 17.0%; A549 – cimetidine treated = 35.5%; ACHN – vehicle treated = 10.3%; ACHN – cimetidine treated = 12.0%), however, the range of RSD among the conditions was more consistent in this replication attempt compared to the original study.

There are multiple approaches that could be taken to explore these data (e.g. MANOVA, regression with RE/AR errors, area under the curve), however to provide a direct comparison to the original analysis, which used volume measurements at day 11, and the available original data, we are reporting the analysis specified *a priori* in the Registered Report (*Kandela et al., 2015*). To test if cimetidine is effective in the A549 derived xenograft model, but not in the ACHN derived xenograft model, we performed an analysis of variance (ANOVA) having two levels of tumors (A549 derived or ACHN derived) and two levels of drug treatment (vehicle or cimetidine). The ANOVA result on tumor volumes at day 11 (natural log-transformed) was not statistically significant for all effects at the 0.05 significance level. The interaction effect, with a sample size powered *a priori* to detect the effect based on the originally reported data, was not statistically significant, $F_{(1,53)} = 3.88$, $p=0.054$. Thus, the null hypothesis that the effect of the drug treatment (difference between vehicle and cimetidine) on inhibiting tumor volume growth is similar for A549 and ACHN derived tumors can not be rejected. The main effect for cell line, $F_{(1,53)} = 1.21$, $p=0.277$, and drug treatment $F_{(1,53)} = 1.13$, $p=0.292$ were also not statistically significant.

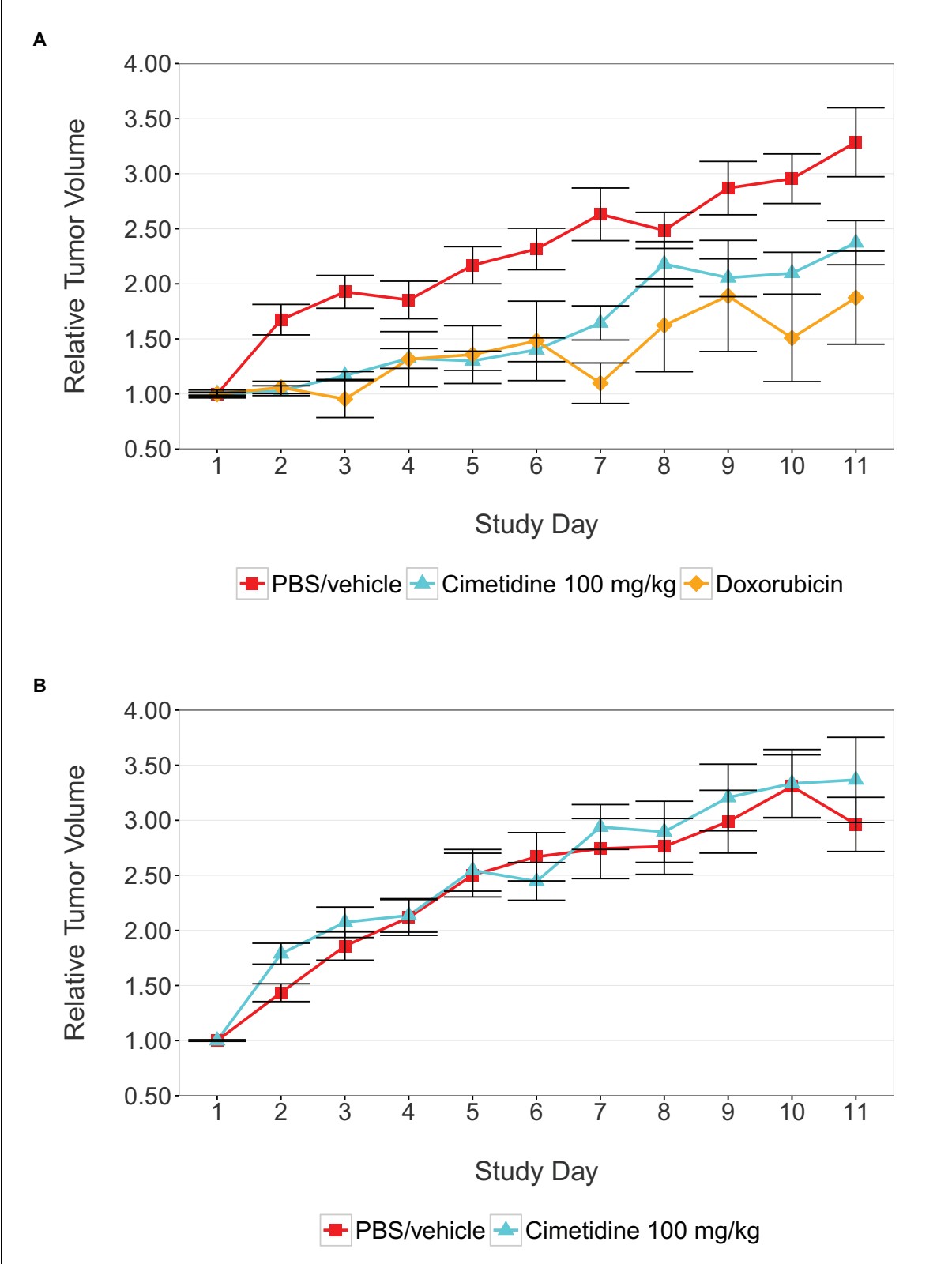

**Figure 1.** Tumor xenograft experiment testing efficacy of cimetidine in inhibiting the growth of tumors in SCID mice. Female SCID mice bearing subcutaneous A549 human lung adenocarcinomas (**A**) or ACHN renal cell carcinomas (**B**) were randomized into three treatment groups once tumors reached a calculated volume of 100 mm$^3$. Mice were intraperitoneally injected with 100 mg/kg cimetidine, vehicle (PBS), or 2 mg/kg doxorubicin. Tumor volumes were measured by caliper for the duration of the study. For each condition tumor volume was normalized relative to the average Day 1 values.
*Figure 1 continued on next page*

*Figure 1 continued*

Means reported and error bars represent s.e.m. Number of mice: (**A**) A549 tumors: n=14 (vehicle), n=13 (cimetidine), n=4 (doxorubicin) (**B**) ACHN tumors: n=15 (vehicle), n=15 (cimetidine). Two-tailed Welch's *t*-test on Day 11 relative volumes between vehicle or doxorubicin treated A549 tumors; *t* (6.73) = 2.68, uncorrected *p*=0.0325 with *a priori* alpha level = 0.0167; (Bonferroni corrected *p*=0.0976). Two-way ANOVA interaction between cell line (A549 or ACHN) and treatment (vehicle or cimetidine) on natural log transformed Day 11 relative volumes; *F*(1,53) = 3.88, *p*=0.054. Pairwise contrast between A549 tumors treated with vehicle or cimetidine; *t*(53) = 2.16, uncorrected *p*=0.035 with *a priori* alpha level = 0.0167; (Bonferroni corrected *p*=0.105). Pairwise contrast between ACHN tumors treated with vehicle or cimetidine; *t*(53) = 0.58, uncorrected *p*=0.562 with *a priori* alpha level = 0.0167; (Bonferroni corrected *p*>0.99). Additional details for this experiment can be found at https://osf.io/fh6gn/.

The following figure supplement is available for figure 1:

**Figure supplement 1.** Individual tumor xenografts.

As outlined in the Registered Report (*Kandela et al., 2015*), we planned to conduct three comparisons using the Bonferroni correction to adjust for multiple comparisons. Although the Bonferroni method is conservative, it was accounted for in the power calculations to ensure sample size was sufficient. The planned pairwise comparison of the tumor volumes from A549 derived tumors treated with vehicle compared to cimetidine was not statistically significant (*t*(53) = 2.16, uncorrected *p*=0.035, *a priori* Bonferroni adjusted significance threshold = 0.0167; (Bonferroni corrected *p*=0.105)). Additionally, the planned pairwise comparison of ACHN derived tumors treated with vehicle compared to cimetidine was not statistically significant (*t*(53) = −0.58, uncorrected *p*=0.562, *a prior* Bonferroni adjusted significance threshold = 0.0167; (Bonferroni corrected *p*>0.99)). While not statistically significant, the treatment of A549 derived tumors with cimetidine resulted in decreased tumor volume compared to vehicle control at day 11, which is in the same direction as the original study. The treatment of ACHN derived tumors with cimetidine did not result in an observational difference in tumor volume after treatment when compared to vehicle control treated tumors, similar to the original study.

As an additional control for tumor growth inhibition, tumors from A549 cells were treated with 2 mg/kg doxorubicin, similar to the original experimental design (*Figure 1A*). While tumors from A549 cells treated with vehicle grew to an average of 3.29 times their original volume by day 11, tumors in mice treated with doxorubicin grew to an average of 1.87 times the original volume by day 11 [n=4, SD=0.845], similar to the ~2.0 times, reported in *Sirota et al., 2011* in the doxorubicin condition. Analysis of vehicle treated compared to doxorubicin treated tumor volumes at day 11, which was planned *a priori*, was not statistically significant (Welch's *t*-test; *t*(6.73) = 2.68, uncorrected *p*=0.0325, *a priori* Bonferroni adjusted significance threshold = 0.0167; (Bonferroni corrected *p*=0.0976)). While not statistically significant, the treatment of tumors from A549 cells with doxorubicin resulted in a smaller volume when compared to vehicle control at day 11, which is in the same direction as the original study. Additionally, mice treated with doxorubicin started to lose weight and had a rough coat starting on approximately day 7 of the treatment schedule. A small reduction in tumor volume compared to vehicle control has also been reported in other studies that utilized low dose doxorubicin experimental designs (*Hossain et al., 2012*; *Lopez et al., 2009*; *Wang et al., 2013*, *2010*).

At the completion of the treatment schedule, tumors were excised and imaged (available at https://osf.io/xcuh6/), similar to the representative A549 derived tumors treated with vehicle or 100 mg/kg cimetidine shown in Figure 4D of the original study (*Sirota et al., 2011*). The excised tumors were also weighed (*Figure 2*), as described in the Registered Report (*Kandela et al., 2015*). This was an additional measured parameter not included in the original study. The tumor weights were subjected to a two-way ANOVA having two levels of cell lines (A549 derived or ACHN derived), two levels of treatment (vehicle or cimetidine), and their interaction. The main effect of cell lines yielded an *F* ratio of *F*(1,53) = 7.560, *p*=0.00814, indicating that the mean tumor weight was significantly larger for A549 derived tumors [n=27, M=0.138, SD=0.0423] than for ACHN derived tumors [n=30, M=0.108, SD=0.0402]. The main effect of treatment was not significant (*F*(1,53) = 0.586, *p*=0.447) nor was the interaction effect (*F*(1,53) = 0.00005, *p*=0.994). Exploratory analysis of vehicle treated [n=14, M=0.134, SD=0.0444] compared to doxorubicin treated [n=4, M=0.110, SD=0.0576] A549 derived tumor weights was not statistically significant (Welch's *t*-test; *t*(4.078) = 0.771, *p*=0.483). While not statistically significant, the treatment of tumors derived from A549 cells with doxorubicin

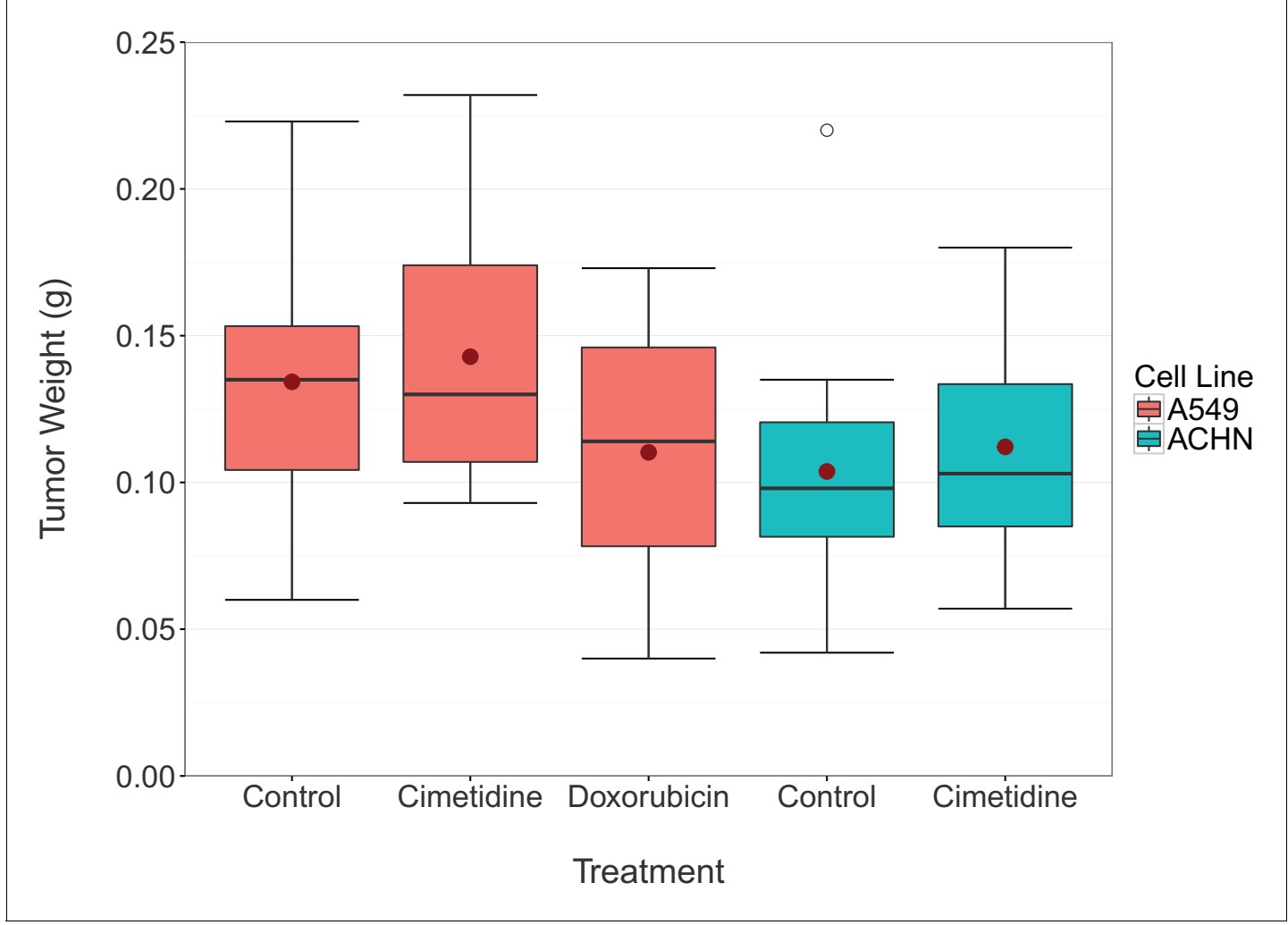

**Figure 2.** Final tumor weights from xenograft experiment testing efficacy of cimetidine in inhibiting the growth of tumors in SCID mice. At the end of the predefined study period (Day 12), tumors from the xenograft experiment reported in *Figure 1* were excised and weighed. Box and whisker plot with median represented as the line through the box, means represented as the solid red circle, and whiskers representing values within 1.5 IQR of the first and third quartile. Number of mice: A549 tumors: n=14 (vehicle), n=13 (cimetidine), n=4 (doxorubicin). ACHN tumors: n=15 (vehicle), n=15 (cimetidine). Two-tailed Welch's *t*-test between vehicle or doxorubicin treated A549 tumors; $t_{(4.078)} = 0.771$, $p=0.483$. Two-way ANOVA interaction between cell line (A549 or ACHN) and treatment (vehicle or cimetidine); $F_{(1,53)} = 0.00005$, $p=0.994$. Two-way ANOVA main effect of cell line (A549 or ACHN); $F_{(1,53)} = 7.56$, $p=0.00814$. Additional details for this experiment can be found at https://osf.io/fh6gn/.

resulted in a small effect (Cohen's $d$ = 0.51 [−0.62, 1.63]), which is in the expected direction. A small reduction in tumor weight compared to vehicle control has also been reported in other studies that utilized a similar low dose doxorubicin experimental design (*Biswas et al., 2013*; *Hossain et al., 2012*; *Lopez et al., 2009*; *Wang et al., 2010*).

## Meta-analyses of original and replicated effects

We performed a meta-analysis using a random-effects model to combine each of the effects described above on day 11 tumor volume comparisons as pre-specified in the confirmatory analysis plan (*Kandela et al., 2015*). To provide a standardized measure of the effect Cohen's $d$ was calculated for the original and replication studies. Cohen's $d$ is the standardized difference between two means using the pooled sample standard deviation.

The comparison of A549 derived tumor volumes treated with vehicle compared to cimetidine resulted in $d$ = 1.36, 95% CI [0.06, 2.60] for the data estimated *a priori* from Figure 4C of the original

study (*Sirota et al., 2011*). This compares to *d* = 0.93, 95% CI [0.12, 1.72] reported in this study. A

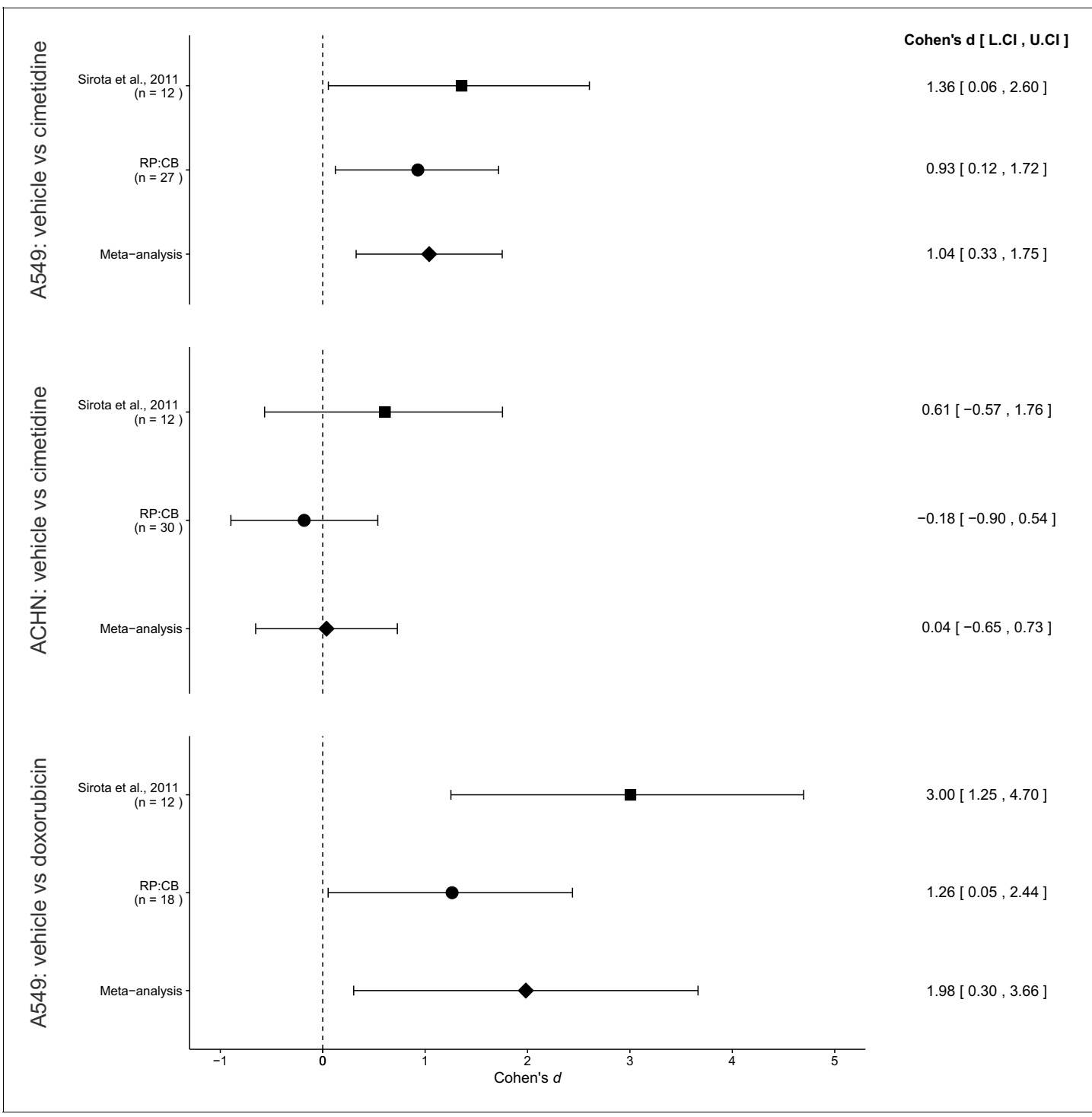

**Figure 3.** Meta-analyses of each effect. Effect size (Cohen's *d*) and 95% confidence interval are presented for *Sirota et al., 2011*, this replication attempt (RP:CB), and a meta-analysis to combine the two effects of day 11 tumor volume comparisons. Sample sizes used in *Sirota et al., 2011* and this replication attempt are reported under the study name. Random effects meta-analysis of A549-derived tumors treated with vehicle (PBS) compared to 100 mg/kg cimetidine (meta-analysis *p*=0.0043), ACHN-derived tumors treated with vehicle (PBS) compared to 100 mg/kg cimetidine (meta-analysis *p*=0.917), and A549-derived tumors treated with vehicle (PBS) compared to 2 mg/kg doxorubicin (meta-analysis *p*=0.021). Additional details for these meta-analyses can be found at https://osf.io/jcghv/.

meta-analysis (*Figure 3*) of these two effects resulted in *d* = 1.04, 95% CI [0.33, 1.75], *p*=0.0043. Both results are consistent when considering the direction of cimetidine's effect. The point estimate of the replication effect size was within the confidence interval of the original result and vice versa. Further, the random effects meta-analysis resulted in a statistically significant effect of cimetidine on inhibiting A549 derived tumors.

The comparison of ACHN derived tumor volumes treated with vehicle compared to cimetidine resulted in *d* = 0.61, 95% CI [−0.57, 1.76] for the data estimated *a priori* from Supplemental Figure 1 of the original study (*Sirota et al., 2011*). This compares to *d* = −0.18, 95% CI [−0.90, 0.54] reported in this study. A meta-analysis (*Figure 3*) of these two effects resulted in *d* = 0.04, 95% CI [−0.65, 0.73], *p*=0.917. The original and replication results were both not statistically significant. While both studies were in opposite directions, the point estimate of the replication effect size was within the confidence interval of the original result, but the original effect size was not within the confidence interval of the replication. The random effects meta-analysis did not result in a statistically significant effect suggesting there is no evidence that cimetidine is effective on ACHN derived tumors.

The comparison of A549 derived tumors treated with vehicle compared to doxorubicin resulted in *d* = 3.00, 95% CI [1.25, 4.70] for the data estimated *a priori* from Figure 4C of the original study (*Sirota et al., 2011*). This compares to *d* = 1.26, 95% CI [0.05, 2.44] reported in this study. A meta-analysis (*Figure 3*) of these two effects resulted in *d* = 1.98, 95% CI [0.30, 3.66], *p*=0.021. Both results are consistent when considering the direction of doxorubicin's effect. The point estimate of the replication effect size was within the confidence interval of the original result, while the point estimate of the original effect size was not within the confidence interval of the replication result. Further, the random effects meta-analysis resulted in a statistically significant effect of doxorubicin on inhibiting A549 derived tumors.

This direct replication provides an opportunity to understand the present evidence of these effects. Any known differences, including reagents and protocol differences, were identified prior to conducting the experimental work and described in the Registered Report (*Kandela et al., 2015*). However, this is limited to what was obtainable from the original paper, which means there might be particular features of the original experimental protocol that could be critical, but unidentified. So while some aspects, such as site of injection, number of cells injected, strain of mice, and manufacturer of the drugs were maintained, others were unknown or not easily controlled for. These include variables such as circadian biological responses to therapy (*Fu and Kettner, 2013*), mouse sex and strain stocks (*Clayton and Collins, 2014*), the microbiome of recipient mice (*Macpherson and McCoy, 2015*), housing temperature in mouse facilities (*Kokolus et al., 2013*), cell line drift (*Hughes et al., 2007*), and differing compound potency resulting from different stock solutions (*Kannt and Wieland, 2016*). Whether these or other factors influence the outcomes of this study is open to hypothesizing and further investigation, which is facilitated by direct replications and transparent reporting.

## Materials and methods

As described in the Registered Report (*Kandela et al., 2015*), we attempted a replication of the experiments reported in Figures 4C–D and Supplemental Figure 1 of *Sirota et al., 2011*. A detailed description of all protocols can be found in the Registered Report (*Kandela et al., 2015*). Additional detailed experimental notes, data, and analysis are available on the Open Science Framework (OSF) (RRID: SCR_003238) (https://osf.io/hxrmm/; *Kandela et al., 2016*).

### Cell culture

A549 cells (ATCC, CCL-185) were maintained in F-12 Ham's medium supplemented with 10% Fetal Bovine Serum (FBS) (Sigma-Aldrich, cat # F2442) and 2 mM sodium pyruvate. ACHN cells (ATCC, CRL-1611) were maintained in EMEM supplemented with 10% FBS, 2 mM glutamine, and 1 mM sodium pyruvate. All cells were grown at 37°C and at 5% $CO_2$. Quality control data for the A549 and ACHN cell lines are available on the OSF (https://osf.io/yt9fv/). This includes results confirming the cell lines were free of mycoplasma contamination and common mouse pathogens. Additionally, STR DNA profiling of the cell lines was performed and all cells were confirmed to be the indicated cell lines when queried against STR profile databases.

## Animals

All animal procedures were approved by the Northwestern University IACUC# IS00000424 and were in accordance with the Northwestern University's policies on the care, welfare, and treatment of laboratory animals. No blinding occurred during the experiments.

Six-week old female SCID mice (Charles River, strain code 236) were inoculated subcutaneously (s.c.) with A549 or ACHN cells at a density of $5 \times 10^6$ cells in 100 μl of Dulbecco's Phosphate Buffered Saline (DPBS) (Sigma-Aldrich, cat # D8537) in the right upper flank. Tumor (caliper measurements) and body weight (scale: Scout Pro O'Haus, model # SP202 (S/N: B443188875)) were measured three times a week before enrollment to reduce stress in the mice and were measured daily during the course of the experiment. Once the tumor reached 100 mm$^3$, the mice were randomized into the treatment groups (vehicle (DPBS), 100 mg/kg cimetidine (dissolved in DPBS and pH adjusted and filtered through 0.22 μm syringe), or 2 mg/kg doxorubicin (dissolved in DPBS and filtered through 0.22 μm syringe)), taking into account the pre-specified group sizes. Two mice were not enrolled in the study due to unanticipated death and a small tumor size. Injections were administered via the intraperitoneal (IP) route using a 1 ml syringe (BD, cat # Z192090) with 27 gauge needles (BD, cat # 305109) as a repeat dose each day for cimetidine and twice a week for doxorubicin for 11 days with a dosing volume of 100 μl per 20 gram mouse. The volume of injection was calculated based on the total body weight obtained on the same day prior to injection. After repeat dosing for 11 days, all surviving animals were sacrificed 24 hr post last dose. Tumors were harvested and weighed (scale: Metler Toledo, Model # XL300), with pictures of the tumors taken with a ruler for scale similar to the representative examples shown in Figure 4D of the original study (available at: https://osf.io/xcuh6/).

Each cage contained up to five mice and offered Certified Rodent Diet (Harlan Teklad, cat # 7912) *ad libitum*. The animal room was set to maintain between 68–75°F, a relative humidity of 30–70%, a minimum of 15 room air changes per hour, and a 12 hr light/dark cycle, which was interrupted for study-related activities.

## Statistical analysis

Statistical analysis was performed with R software (RRID: SCR_001905), version 3.2.3 (*R Core Team, 2016*). All data, csv files, and analysis scripts are available on the OSF (https://osf.io/hxrmm/). Confirmatory statistical analysis was pre-registered (https://osf.io/yc8k2/) before the experimental work began as outlined in the Registered Report (*Kandela et al., 2015*). Additional exploratory analysis was performed using the excised tumor weights. Data were checked to ensure assumptions of statistical tests were met. A meta-analysis of a common original and replication effect size was performed with a random effects model and the *metafor* package (*Viechtbauer, 2010*) (available at: https://osf.io/jcghv/). The original study data was extracted *a priori* from the published figures by determining the mean and upper/lower error values for each data point. The extracted data was published in the Registered Report (*Kandela et al., 2015*) and was used in the power calculations to determine the sample size for this study.

## Deviations from registered report

The source of FBS and DPBS were different than what is listed in the Registered Report, with the used source and catalog number listed above. (note: the original source was not specified). Additional materials and instrumentation not listed in the Registered Report, but needed during experimentation are also listed.

## Acknowledgements

The Reproducibility Project: Cancer Biology would like to thank the following companies for generously donating reagents to the Reproducibility Project: Cancer Biology; American Type and Tissue Collection (ATCC), Applied Biological Materials, BioLegend, Charles River Laboratories, Corning Incorporated, DDC Medical, EMD Millipore, Harlan Laboratories, LI-COR Biosciences, Mirus Bio, Novus Biologicals, Sigma-Aldrich, and System Biosciences (SBI).

# Additional information

## Group author details

Reproducibility Project: Cancer Biology

Elizabeth Iorns: Science Exchange, Palo Alto, United States; Stephen R Williams: Center for Open Science, Charlottesville, United States; Nicole Perfito: Science Exchange, Palo Alto, United States; Timothy M Errington, http://orcid.org/0000-0002-4959-5143: Center for Open Science, Charlottesville, United States

## Competing interests

IK and FA: Developmental Therapeutics Core is a Science Exchange associated lab. RP:CB: EI, NP: Employed by and hold shares in Science Exchange Inc.

## Funding

| Funder | Author |
|---|---|
| Laura and John Arnold Foundation | Reproducibility Project: Cancer Biology |

The funder had no role in study design, data collection and interpretation, or the decision to submit the work for publication.

## Author contributions

IK, FA, Acquisition of data, Drafting or revising the article; RP:CB, Analysis and interpretation of data, Drafting or revising the article

## Ethics

Animal experimentation: All animal procedures were approved by the Northwestern University IACUC# IS00000424 and were in accordance with the Northwestern University's policies on the care, welfare, and treatment of laboratory animals.

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
