## [Decision Letter]

Thank you for submitting your work entitled "Replication Study: Discovery and Preclinical Validation of Drug Indications using Compendia of Public Gene Expression Data" for further consideration at *eLife*. Your article has been evaluated by Charles Sawyers as the Senior Editor, a Reviewing Editor, and three reviewers.

There are issues that need to be addressed in a resubmission, as outlined below:

In this manuscript, Kandela and colleagues attempt to replicate prior work by Sirota et al. on the validation of computationally predicted sensitivity to the drug cimetidine in lung but not in renal cancer xenografts. They conclude that the sensitivity to cimetidine of xenografts from A549 lung cancer cells, compared to media control cannot be assessed at a statistically significant level. The authors' results show a trend of response to cimetidine as the original findings but possibly less potent (less significant).

1) It is notable that there are variables such as: circadian biological responses to therapy, mouse strain stocks, and the microbiome of recipient mice that may affect the growth of xenografts. As such, it is suggested that the authors provide a discussion that includes variables that were not specifically addressed or could not be easily controlled.

2) The original study observed a statistically significant reduction in A549 tumor volume while the current study did not, although the direction and magnitude of changes are similar. The number of mice in the study was inferred based on the original study, based on smaller tumor volume measurement errors. The current study conducted pre-planned contrasts on log transformed data within the framework of ANOVA, and the p-values are Bonferroni corrected, while the previous study performed t-tests on un-transformed data without Bonferroni correction. Why is Bonferroni correction used in the new study as opposed to testing directly the single original hypothesis, i.e. that tumor growth of cimetidine treated mice at the highest concentration is reduced compared to PBS/vehicle treatment? We don't see the rationale here for introducing a Bonferroni correction. In addition, it is well known that Bonferroni is an ultra-conservative way to account for multiple hypothesis testing. Without Bonferroni correction, the p-value is significant (p = 0.035) despite the larger error size. Please explain and discuss.

3) Please note that there are more effective models in which the curves can be easily fit by regression analysis (e.g. MANOVA or Regression with RE/AR errors). Such models could be used here. Using the last time point is especially sensitive to measurement errors. It is evident by comparing Figure 1 to the original plot that the measurement errors incurred by the reproducibility study are substantially larger than those in the original study (Figure 4C). This is a problem also because the same errors were substantially reduced when comparing cimetidine activity to PBS/vehicle in ACHN cells (Figure 1). This raises the concern regarding the accuracy of caliper measurements and tumor volume assessment by Kandela et al. Please explain these issues and provide the data plotted as individual xenografts rather than averages in the supplemental data so that readers can examine the extent of mouse to mouse variations.

4) Please address the following minor issues. There are substantial variability sources in the study that may cause small effect changes. These include:

a) Drift in the A549 cell line: was this authenticated in the repeated study compared to the original one?

b) Different compound potency from different stock solutions: were titration curves performed to assess whether the EC50 of the compounds recapitulated those in the original study? This is especially critical since the growth curves at 50mg/kg and 100mg/kg were dramatically different. Thus even a minimal difference in compound potency could induce profound differences in the measurements.

c) Site of injection: Were the sites identical?

---

## [Author Response]

*Thank you for submitting your work entitled "Replication Study: Discovery and Preclinical Validation of Drug Indications using Compendia of Public Gene Expression Data" for further consideration at eLife. Your article has been evaluated by Charles Sawyers as the Senior Editor, a Reviewing Editor, and three reviewers.*

*There are issues that need to be addressed in a resubmission, as outlined below:*

*In this manuscript, Kandela and colleagues attempt to replicate prior work by Sirota et al. on the validation of computationally predicted sensitivity to the drug cimetidine in lung but not in renal cancer xenografts. They conclude that the sensitivity to cimetidine of xenografts from A549 lung cancer cells, compared to media control cannot be assessed at a statistically significant level. The authors' results show a trend of response to cimetidine as the original findings but possibly less potent (less significant).*

*1) It is notable that there are variables such as: circadian biological responses to therapy, mouse strain stocks, and the microbiome of recipient mice that may affect the growth of xenografts. As such, it is suggested that the authors provide a discussion that includes variables that were not specifically addressed or could not be easily controlled.*

Thank you for this comment. We agree and have added a paragraph at the end of the Results/Discussion section to comment on this aspect of conducting replications.

*2) The original study observed a statistically significant reduction in A549 tumor volume while the current study did not, although the direction and magnitude of changes are similar. The number of mice in the study was inferred based on the original study, based on smaller tumor volume measurement errors. The current study conducted pre-planned contrasts on log transformed data within the framework of ANOVA, and the p-values are Bonferroni corrected, while the previous study performed t-tests on un-transformed data without Bonferroni correction. Why is Bonferroni correction used in the new study as opposed to testing directly the single original hypothesis, i.e. that tumor growth of cimetidine treated mice at the highest concentration is reduced compared to PBS/vehicle treatment? We don't see the rationale here for introducing a Bonferroni correction. In addition, it is well known that Bonferroni is an ultra-conservative way to account for multiple hypothesis testing. Without Bonferroni correction, the p-value is significant (p = 0.035) despite the larger error size. Please explain and discuss.*

We performed Bonferroni correction because of the multiple comparisons (3 in total) that we performed on this experimental design. While this is a more conservative approach to adjust for multiple testing, we accounted for this in our power calculations to ensure the sample size was sufficient. The 2x2 design and the multiple planned contrasts was conducted to test the interaction, which the original study implies when they conclude that a statistically significant difference between vehicle and cimetidine in A549 and no difference between vehicle and cimetidine in ACHN is concordant with their computational prediction. However, the difference between significant and not-significant is not necessarily statistically significant (see: Gelman and Stern, 2006; Nieuwenhuis et al., 2011), which is why the replication study designed and performed the statistical tests this way.

To allow readers to interpret the test results, we report the corrected and uncorrected p values. To further clarify why we performed the ANOVA and multiple comparison corrections we revised the manuscript to explain the analysis and how we accounted for the Bonferroni approach in our sample size calculations.

*3) Please note that there are more effective models in which the curves can be easily fit by regression analysis (e.g. MANOVA or Regression with RE/AR errors). Such models could be used here. Using the last time point is especially sensitive to measurement errors. It is evident by comparing Figure 1 to the original plot that the measurement errors incurred by the reproducibility study are substantially larger than those in the original study (Figure 4C). This is a problem also because the same errors were substantially reduced when comparing cimetidine activity to PBS/vehicle in ACHN cells (Figure 1). This raises the concern regarding the accuracy of caliper measurements and tumor volume assessment by Kandela et al. Please explain these issues and provide the data plotted as individual xenografts rather than averages in the supplemental data so that readers can examine the extent of mouse to mouse variations.*

Thank you for this suggestion. We agree that different models to analyze the data could be used and we encourage others to explore the data this way. This is one of the reasons we are making our raw data available. However, for this specific project we have restricted our analysis to what we specified in the Registered Report and how the sample size was determined. The final caliper measurement was also the approach taken in the original paper. Also, during peer review of the Registered Report we proposed performing an exploratory analysis of all the data, but removed this as suggested by a reviewer since in vivo caliper measurements are noisy, especially with the earlier time-points when the tumors are smaller in size. We have revised the manuscript to communicate how multiple approaches could be taken, but we have limited our analysis to what we proposed.

We have also revised the manuscript to state the error of the measurements between the mice in this replication attempt, which generally had larger relative standard deviations compared to the original study. However, this replication was more consistent among the conditions compared to the original study. (Replication attempt: (A549 – vehicle treated = 35.6%; A549 – cimetidine treated = 30.5%; ACHN – vehicle treated = 32.1%; ACHN – cimetidine treated = 44.5%). Original study: (A549 – vehicle treated = 17.0%; A549 – cimetidine treated = 35.5%; ACHN – vehicle treated = 10.3%; ACHN – cimetidine treated = 12%)). Additionally, we have included a supplement figure for Figure 1, (Figure 1—figure supplement 1), which plots the data for each individual animal rather than the averages.

*4) Please address the following minor issues. There are substantial variability sources in the study that may cause small effect changes. These include:*

*a) Drift in the A549 cell line: was this authenticated in the repeated study compared to the original one?*

We confirmed the A549 and ACHN cells were correct by STR profile and free of mycoplasma and rodent pathogens. However, we do not know the profile of the original cell lines as these were not shared, nor was the source of the cell lines identified in the original study. We have included this aspect as a feature that might be critical, but unidentified at the end of the Results/Discussion section.

*b) Different compound potency from different stock solutions: were titration curves performed to assess whether the EC50 of the compounds recapitulated those in the original study? This is especially critical since the growth curves at 50mg/kg and 100mg/kg were dramatically different. Thus even a minimal difference in compound potency could induce profound differences in the measurements.*

We did not perform titration curves in this replication. We agree that slight differences in compound potency can affect the outcome of the experiment. We have included this aspect as a feature that might be critical, but not controlled for at the end of the Results/Discussion section.

*c) Site of injection: Were the sites identical?*

The injection site in the original paper was reported as occurring in the upper flank of the animals and intraperitoneally. This was the same for the replication attempt.